# Creep Mechanisms of an Al–Cu–Mg Alloy at the Macro- and Micro-Scale: Effect of the S′/S Precipitate

**DOI:** 10.3390/ma12182907

**Published:** 2019-09-09

**Authors:** Yongqian Xu, Lingwei Yang, Lihua Zhan, Hailiang Yu, Minghui Huang

**Affiliations:** 1State Key Laboratory of High-Performance Complex Manufacturing, Central South University, Changsha 410083, China (Y.X.) (L.Y.) (H.Y.) (M.H.); 2College of Mechanical and Electrical Engineering, Central South University, Changsha 410083, China; 3Light Alloy Research Institute, Central South University, Changsha 410083, China; 4China Aerodynamics Research and Development Center, Mianyang 621000, China

**Keywords:** creep age forming, creep behavior, precipitate, nano-indentation, Al–Cu–Mg alloy

## Abstract

A novel methodology combining the macro- and micro-creep techniques was employed to study the effect of S′/S precipitate growth on the creep mechanism of an Al–Cu–Mg alloy. An AA2524 alloy was pre-aged at 180 °C to obtain S′/S precipitates with various sizes. The results showed that the precipitate size increased approximately linearly to ≈32 nm, ≈60 nm, and ≈105 nm after 3 h, 6 h, and 12 h of pre-aging, respectively. The growth of precipitate could significantly shorten the primary creep stage, despite the fact that the steady-state creep behavior was similar to that of the as-received alloy, as revealed by the macro tensile creep tests at 180 °C and 180 MPa. This led to a stress exponent (2.4–2.5) of the Al alloy with various precipitate sizes that was quite close to that of the as-received Al alloy, implying a steady-state creep mechanism dominated by grain boundary sliding and dislocation interactions. Finally, the micro-creep tests showed a minor role of the precipitate size on the steady-state creep mechanism, as evidenced by the similar strain rate sensitivity (0.0169–0.0186), activation volume (≈27 b^3^), and the results of a detailed transmission electron microscopy analysis of all tested alloys.

## 1. Introduction

Al–Cu–Mg alloys (known as 2000 series) have been widely applied in aircraft components due to their low density and high damage tolerance [1,2,3]. Some complex aircraft components are generally manufactured by creep age forming (CAF), which is a new processing technique to simultaneously strengthen the Al–Cu–Mg alloys and change their shapes during a one-step forming and heat-treatment process [4,5,6]. In order to acquire aircraft components with high performance and high precision by CAF, it is vitally important to study fundamentally the creep aging mechanism of Al–Cu–Mg alloys. Macroscale creep tests have been widely employed to evaluate the creep properties of Al alloys by correlating the processing conditions (creep stress, temperature, and aging time) with the microstructures (precipitate, grain size, grain boundaries, etc.) and creep properties [7,8,9,10,11,12]. Zhan et al. [10] explored the microstructures and mechanical properties of creep-aged Al–Cu–Mg alloy sheets (AA2524) depending on creep temperature, aging time, and sheet thickness. Maximov et al. [11] quantified the effect of creep temperature on strain hardening and creep behavior of an Al–Cu–Mg alloy (2024-T3). Chen et al. [12] investigated the evolution of S′ precipitate in an Al–Cu–Mg alloy during stress-free and stress aging. They found that the precipitation distribution of the S′ phase during stress aging could be changed by the loading orientation of the applied stress. Moreover, compressive stress aging may lead to a shorter S′ phase, and the length of the S′ phase tends to decrease with the increase of the applied stress. However, the effect of the growth of S′/S precipitates on the creep mechanism has not been reported yet.

Since the macro-creep behavior of typical Al alloys generally combines contributions from multiscale factors, such as nanoscale precipitates, microscale grains, meso- and macroscale porosities, it is still challenging to decouple the effect of each microstructural feature and to rationalize the creep mechanism of typical Al–Cu–Mg alloys for CAF applications. In order to highlight the contribution of individual microstructures on the creep mechanism of a material, microscale creep techniques based on the indentation technique [13,14,15,16] have been developed recently, such as constant strain rate, constant load, indentation strain-rate jump techniques. Among them, the indentation strain-rate jump technique is advantageous to acquire rate-dependent strength during single tests and has been applied in multiple metal/alloy systems (nanograined Ni, ultrafine-grained Al, etc. [17,18,19,20]). However, this technique has not been extensively applied to Al–Cu–Mg alloys to study the basic creep mechanisms dominated by particular microstructures such as precipitates during CAF. 

This study focuses on the investigation of the creep mechanism of an Al–Cu–Mg alloy, exploring the effect of precipitate size by a combined micro- and macro-creep technique. A typical AA2524 alloy was pre-aged at high temperature (180 °C) to obtain precipitates with tailored size. Transmission electron microscopy (TEM) was used to study the precipitate microstructures and to measure the precipitate sizes after various pre-aging times. The effect of the precipitate size on the primary and steady-state creep behaviors of the Al alloy was studied by macro-creep tests, and the dominated macro-creep mechanism was discussed. In order to clarify the role of the precipitate size on the creep mechanism at the microscale, indentation strain-rate jump tests were performed on the Al alloy with different precipitate sizes. The rate-dependent creep was discussed with the help of the measured strain rate sensitivity and activation volume of the Al alloy. The creep mechanism was finally verified by direct observations of the interactions of dislocations with the precipitates during micro-creeping by TEM. 

## 2. Material and Methods

A commercial high-strength cold-rolled AA2524 alloy was used in this study. The as-received alloy has undergone T3 treatment, i.e., 1–2% pre-deformation at room temperature after solution treatment to eliminate quenching residual stress. Its chemical composition is presented in Table 1. Prior to the creep tests, the as-received alloy was pre-aged for 3 h, 6 h, and 12 h at 180 °C, in order to obtain precipitates with different microstructures. An optical microscope Olympus IX71 (Olympus, Tokyo, Japan) was employed to characterize the grain structures of the Al alloy after pre-aging processing. In addition, TEM (Tecnai-F20, FEI, Hillsboro, OR, USA) was employed to study the evolution of the precipitate with pre-aging time. To achieve this, TEM lamellae were fabricated by electro-polishing of a mechanically thinned disc (3 mm in diameter and 60–80 μm in thickness). After polishing, the total thicknesses of all TEM lamellae were <100nm, and nanoscale precipitates could be characterized. 

### 2.1. Macro-Creep Tests

A constant stress tensile creep technique was applied to quantify the macro-creep behavior of the Al alloy on a SUST-D5 creep testing machine (SUST, Zhuhai, China) with an assisting furnace. The creep specimen with gauge length of 50 mm was machined by wire-electrode cutting along the rolling direction of the AA2524 alloy sheet [3]. The detailed dimensions of the creep specimen are shown in Figure 1. The creep temperature was 180 °C to obtain an Al alloy with high strength and fine precipitates. In order to measure the temperature of the specimen precisely, three thermocouples (K-type; Ni/Al-Ni/Cr) were tied on the top, middle, and bottom of the specimen. The creep stress was 180 MPa, and the creep time was 12 h. During the creep tests, the creep load was applied after the temperature of the specimen reached a steady value of 180 °C at a heating rate of 5 °C/min. At the end of the tests, the applied loading was released, and the specimen was naturally cooled down to room temperature outside the furnace. Uniaxial tensile tests were also performed at macroscales using an SUST-CMT5105 machine (SUST, Zhuhai, China) at room temperature with a strain rate of 0.033 mm/s. Each experiment was repeated for at least three times. 

### 2.2. Micro-Creep Tests

A novel indentation strain-rate jump technique was employed to study the creep behavior of the Al alloy at the microscale, following the Maier’s method [17,21]. Prior to the test, the Al alloy was wire-cut (8 × 8 mm^2^) and finely polished by diamond particles (<1 μm) to eliminate the artificial residual stress. The indentation test was implemented in an Agilent G200 nanomechanical system (Keysight, Santa Rosa, CA, USA) with a Berkovich diamond indenter. A continuous stiffness method (CSM) was used to insert the indenter on the finely polished sample to a fixed depth, e.g., 2500 nm in this work, and then unload to recover the elastic deformation. The shallow penetration depth can impose a stress field that is mainly confined in the grain closest to the indenter, thus the measured indentation is strongly localized. The CSM method is advantageous to acquire the hardness of an alloy continuously, based on the measured indentation force–depth curve, as typically shown in Figure 2a,b. At a shallow indentation depth (<100 nm), the measured hardness was unrealistically high (up to 6 GPa) due to the indentation size effect (ISE). The ISE arises when intense plastic deformation is forced to occur across a very small volume of an initially defect-free crystalline material. As the indenter penetrated deeper, the size effect was weakened, and when the depth surpassed 1500 nm, the hardness stabilized and was almost independent of the indentation depth. A strain-rate jump sequence was thus set at this indentation depth regime, jumping from 0.005/s, to 0.05/s, and to 0.005/s every 250 nm, as depicted in Figure 2a. Each rate jump correlated with a hardness value that was dependent on it (Figure 2b). Thus, the indentation strain-rate jump test is highly efficient to obtain the strength (hardness) of a material at different strain rates during a single indentation test. Since indentation is sensitive to the thermal drift caused by temperature gradients between the diamond indenter and the testing Al alloy, prior to each indentation test, the instrumental stability was assured by bringing the indenter in contact with the specimen surface with a small load (2 μN) and allowing over an hour to reach equilibrium. This method enables a drift rate <0.01 nm/s. In order to acquire reliable indentation results, at least 15 indentation tests were performed, and in each test the indentation was separated by at least 25 μm (several times larger than the average indent width) from that of the previous test. More than two repetitive tests were conducted per condition. After the tests, several TEM lamellaes were extracted directly from the imprint cross sections of the Al alloys by the focused ion beam (FIB) technique in a Helios Nanolab G3 UC FIB/SEM dual-beam system (FEI, Hillsboro, OR, USA) for further characterizations. 

## 3. Results and Discussion

### 3.1. Microstructure of the Pre-Aged Al Alloy

Figure 3a,b represent the grain structures of the as-received Al alloy and that after 12 h pre-aging. In addition, grain size statistics was carried out using the Image-Pro software(v6.0, Media Cybernetics, Washington, WA, USA). More than 2000 grains were measured for the 0 h and 12 h pre-aged AA2524. The grains in the as-received Al alloy were equiaxed, and the average grain size was ≈52 μm. The grain structure and size (≈50 μm) were not evidently altered after 12 h pre-aging, evidencing a negligible effect of the pre-aging time. This was expected, since the aging temperature (180 °C) was much lower than the recrystallization temperature of the Al alloy (300 °C) [22].

Figure 4 shows the precipitate microstructures of the Al alloy as a function of pre-aging time. The as-received alloy had a low dislocation density and was free of aging precipitate in individual grains. This is a typical microstructure of commercial AA2524 alloys. Note that pre-existing dislocations can be sources of creep deformation and nucleation sites of aging precipitates [23]. After pre-aging for 3 h at 180 °C, the microstructure was altered by the formation of a large amount of nano-sized precipitate, as shown in Figure 4b. The AA2x24 series alloys are strengthened by the formation of this rod-shaped precipitate during ageing. The precipitation sequence for the ageing of Al–Cu–Mg alloys is: SSS→GPB/S″→S′→S [24,25]. The optimum aging temperature range of AA2524 is 180–190 °C, resulting in fine and uniformly distributed S′/S phases. As it can be seen from Figure 4b, these precipitates were nucleated and grew near the pre-existing dislocations, corresponding to the S′/S phase, and in the form of laths with the {120}_Al_ habit planes elongated along the <100>_Al_ direction [12,26]. They were distributed randomly and uniformly in the Al matrices, without obvious pre-orientation effects. The average size of the precipitate was ≈32 nm. As the pre-aging time increased to 6h and 12h, the size of the precipitate constantly increased to ≈60 nm and ≈105 nm, as shown in Figure 4c,d).

### 3.2. Macro-Creep Mechanism

The growth of the aging precipitates can have a strong effect on the tensile and creep properties of an Al alloy. Figure 5a shows the representative tensile stress–strain curves of the Al alloy as a function of the pre-aging time, showing higher yield strength and tensile strength (max. stress) for longer pre-aging times. The strengthening effect was expected due to the precipitation hardening of the larger precipitate that effectively prohibited dislocation activities. Note such strengthening would lead to degraded fatigue and toughness properties, as indicated by the declined elongation of the Al alloy for longer aging times. This is reasonable, since for typical Al alloys, their strengthening is generally achieved by sacrificing their toughness properties. Table 2 summarizes the tensile property evolution of the Al alloy. It is worthy pointing out that the growth of the aging precipitates can further restrict the creep behavior of the Al alloy, as shown in Figure 3b. After creeping for 12 h at 180 MPa and 180 °C, the ultimate creep strain decreased from 0.179% for the as-received Al alloy to 0.119% after 12 h pre-aging. The decrease of the ultimate creep strain was mainly a consequence of the shortened primary creep stage of the Al alloy for longer pre-aging times. This can be explained by two possible mechanisms: (1) enhanced pinning effect of the dislocation motion by larger precipitates, and (2) decreased vacancies occupied by larger precipitates, which obstructed effectively the atomic diffusion. Following the primary creep, a steady creep stage occurred where the creep rates of the four Al alloys were quite close, i.e., 1.9 × 10^−6^/s, suggesting that the precipitate size had no effect on the steady creep behavior of the Al alloy. 

In order to explore the mechanism dominating the insensitivity of the steady-state creep of the Al alloy with different precipitate sizes, the stress component (*n*) of the as-received Al alloy and after 12 h pre-aging was calculated. The stress exponent [2,9] correlates with the steady-state creep rates (${\dot{\varepsilon }}_{ss}\;$) and the stress levels (*σ*) according to
(1)$$n=\frac{\partial ln{\dot{\varepsilon }}_{ss}}{\partial ln\sigma }$$

In order to calculate *n*, creep tests of the as-received alloy and after 12 h pre-aging were performed under 140 MPa, 180 MPa, and 210 MPa at 180 ℃. Figure 6a,b shows the representative creep strain versus creep time of the Al alloy at different stress levels. As expected, a higher creep strain was triggered at higher stress levels, and pre-aging of the Al alloy hindered effectively its creep response. The ${\dot{\varepsilon }}_{ss}$ was thus plotted as a function of σ in logarithmic coordinates, as shown in Figure 7. Based on Equation (1), the stress exponent was calculated by linear fitting of the data. For the as-received Al alloy, *n* = 2.5 ± 0.4. After 12 h pre-aging, *n* = 2.41 ± 0.6. Considering the error tolerance with heavy n-value overlap, the result suggests strongly that the n exponent is independent of the pre-aging treatments. It has been widely accepted that the value of the stress exponent can be used to indicate the deformation mechanism of steady-state creep at the macroscale [2,27]. When *n* = 1, the creep deformation is mainly controlled by diffusion, which is called diffusion creep. Grain boundary sliding leads to a value of *n* close to 2. The value of *n* = 3–4 is related to dislocation slip mechanism. Based on this, the creep deformations of the Al alloy and after pre-aging were mainly dominated by the combination of grain boundary sliding and dislocation slipping. It seems that the growth of precipitates after pre-aging did not alter the steady creep mechanism of the Al alloy, and this also agrees well with the creep response of Al alloys with various pre-aging time (Figure 5b).

### 3.3. Micro-Creep Mechanism

Based on the aforementioned studies, the steady-state creep of the Al alloy was mainly dominated by grain boundary sliding and dislocation activities in individual grains and seemed independent of the precipitate size. In order to minimize the effect of grain boundaries and explore the steady-state creep mechanism dominated by precipitate growth, an indentation strain-rate jump test was performed on the as-received Al and after 3 h, 6 h, and 12 h of pre-aging. As the penetration depth was only 2500 nm, which covered only a few grains, the main creep strain should be mainly accommodated by the plasticity of the grain closest to the indenter, and this would highlight the effect of the dislocations and their interactions on the precipitate. This was shown by the detailed TEM characterization of the indent imprint cross sections of the as-received Al alloy and of the alloy after 12 h of pre-aging after unloading. In Figure 8a–d, the deformation of the Al alloys under the indenter was highly localized, as evidenced by the high dislocation densities in the grains closest to the indenter. Due to the inhomogeneous stress field generated, shear bands were generated where the stress was concentrated in the grain. Massive dislocation activities, such as dislocation entangles, jogs, dislocation slips, were observed, which suggests a major role of dislocation in the indentation creep mechanism of the Al alloys under study. 

Figure 9a plots the indentation hardness versus the indentation depth during indentation strain-rate jumping from 0.05/s to 0.0005/s, showing variations of the hardness at different strain rates. In agreement with the macro-tensile results (Figure 5a), the micro hardness of the Al alloy increased as the pre-aging time increased from 0 h to 12 h, due to the precipitate hardening effect, as shown in Figure 9b. It is instructive that a higher strain rate generally yielded a stronger response for all alloys, which indicates an apparent time-dependent (creep) property of the Al alloy at the micro scale.

Based on the results in Figure 9, the strain rate sensitivity (m [28,29], defined as the logarithmic changes in stress σ divided by the strain rate $\dot{\varepsilon }$ of the Al alloys) could be calculated by:(2)$$m=\frac{\partial \ln\sigma }{\partial \ln\dot{\varepsilon }}\approx \frac{\partial \mathrm{lnH}}{\partial \ln\dot{\varepsilon }}$$

Table 3 lists the strain rate sensitivities and activation volumes of the Al alloys with different pre-aging times. Also, the average and standard deviation of m (0.0177 ± 0.0008) and V* (≈27.36 ± 0.64 b^3^) of four different preaged samples are reported in Table 3. The low standard deviation of the data indicates that the strain rate sensitivities and activation volumes remained almost constant. The experimental results showed that the changes of m and V* with nucleation and growth of S/S′ precipitates were not obvious, because the micro-creep mechanism was mainly caused by dislocation interactions (TEM images shown in Figure 8). Within the error tolerance, m for all alloys was ranged 0.0169–0.0186, in good agreement with that of typical Al alloys [20,30,31,32]. This result also correlates well with the similar steady-state creep response of AA2524 alloys (Figure 5b), whose steady state creep rate is insensitive to the aging precipitation process. The strain rate sensitivity values also imply a mechanism dominated by dislocation interactions during micro-creep deformation of the Al alloys [33]. This explains well the lack of effect of precipitate growth on the indentation creep mechanism and macro steady-state creep mechanism.

The activation volume (V*), defined as the separation distance between points of dislocation intersection [34], is a direct measure of the deformation mechanism during indentation creep and can be correlated with the strain rate sensitivity by:(3)$$V^{*}=\sqrt{3}\times K_{B}T\times \left(\frac{\partial \ln\dot{\varepsilon }}{\partial \sigma }\right)\approx \frac{3\sqrt{3}\times K_{B}T}{m\times H}$$

Here, K_B_ is the Boltzmann constant, and T is the absolute temperature during indentation creep (≈300 K). The calculated V* for all alloys are also listed in Table 3. They ranged between 26.67 and 28.18, within error tolerance. The values strongly suggest a creep mechanism dominated by dislocation interactions (thermally activated dislocations) that have been found in other Al alloys [21,35], in accordance with the mechanisms implied by the m values for the Al alloys. 

## 4. Conclusions

(1) A novel methodology combining macro- and micro-creep techniques was employed to study the effect of S′/S precipitate growth on the creep mechanism of a typical Al–Cu–Mg alloy (AA2524). 

(2) High-density S′/S precipitates were generated in the Al alloy after pre-aging at 180 °C, and the precipitate size increased approximately linearly to ≈32 nm, ≈60 nm, and ≈105 nm after 3 h, 6 h, and 12 h of pre-aging, respectively. Despite this, the grain structure was not altered. 

(3) The precipitate growth could strengthen effectively the Al alloy, but the creep behavior was suppressed. The macro-creep tests revealed that the precipitate growth could significantly shorten the primary creep time, mainly due to the precipitates pinning dislocation. However, the steady-state creep rate was approximately unchanged. The calculated stress exponent was 2.4–2.5, implying a macro steady creep mechanism dominated by grain boundary sliding and dislocation slipping. 

(4) To eliminate the effect of grain boundaries and to highlight the role of the precipitates, micro strain-rate jump tests were applied. The results suggest a minor role of the S′/S precipitates on the creep mechanism of the Al alloys, since the dominant mechanism was still controlled by dislocations rather than by the aging precipitates.

## Figures and Tables

**Figure 1 materials-12-02907-f001:**
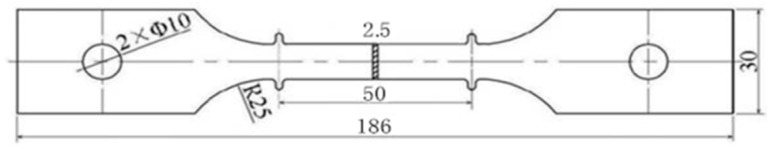
Geometry and dimensions of the creep specimen (unit: mm).

**Figure 2 materials-12-02907-f002:**
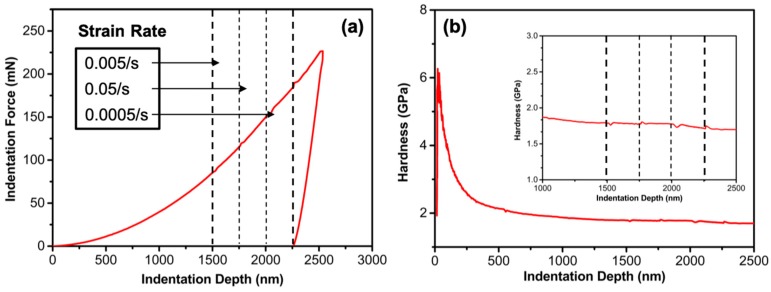
(**a**) Representative indentation force–depth curve of a strain-rate jump test. (**b**) Measured hardness versus indentation depth curve. The insert is a magnified figure of the strain-rate jump regime. The changes in the hardness are highlighted.

**Figure 3 materials-12-02907-f003:**
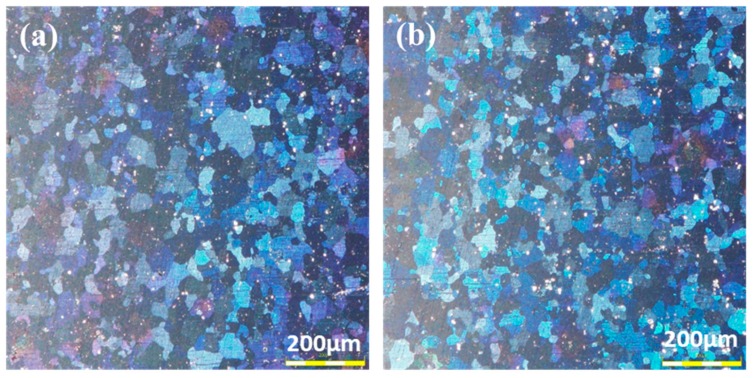
Optical micrograph illustrating the similar equiaxed grain structure of (**a**) 0 h and (**b**) 12 h pre-aged AA2524 alloy.

**Figure 4 materials-12-02907-f004:**
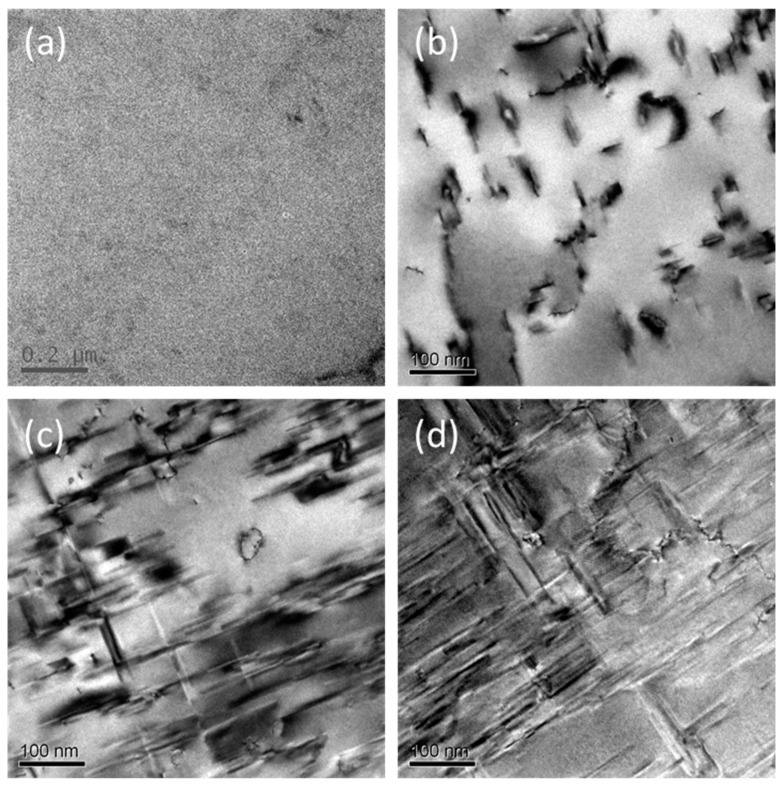
TEM images of intragranular microstructures of the pre-aged AA2524 alloy. (**a**) As-received material; (**b**) 3 h pre-aged; (**c**) 6 h pre-aged, and (**d**) 12 h pre-aged at 180°C.

**Figure 5 materials-12-02907-f005:**
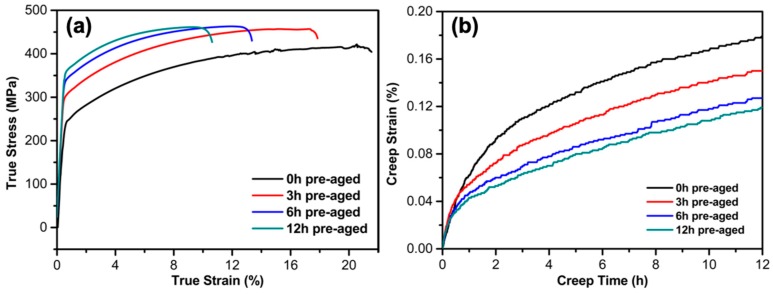
(**a**) Tensile stress–strain curves of the Al alloy as a function of pre-aging time; (**b**) creep strain–time curves of the alloy as a function of pre-aging time (elastic strain was removed).

**Figure 6 materials-12-02907-f006:**
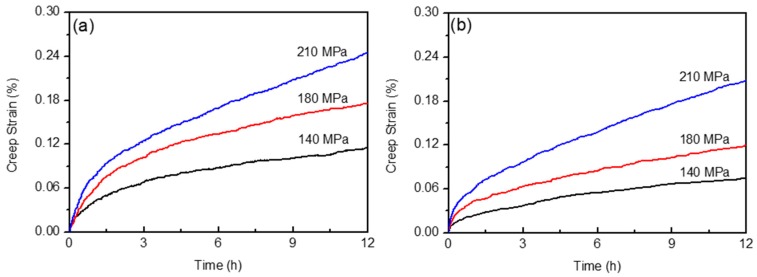
Creep behaviors of the 0 h- (**a**) and 12 h- (**b**) preaged AA2524 under the applied stress of 140 MPa, 180 MPa, and 210 MPa.

**Figure 7 materials-12-02907-f007:**
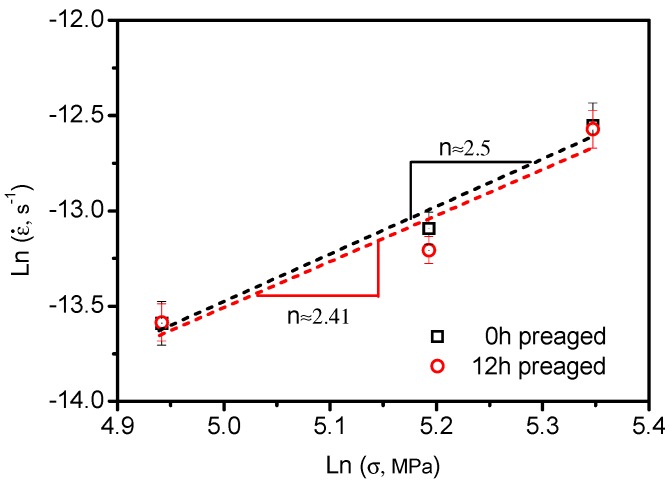
Relationship between lnσ and $\ln{\dot{\varepsilon }}_{ss}$ of the 0 h- and 12 h-preaged AA2524.

**Figure 8 materials-12-02907-f008:**
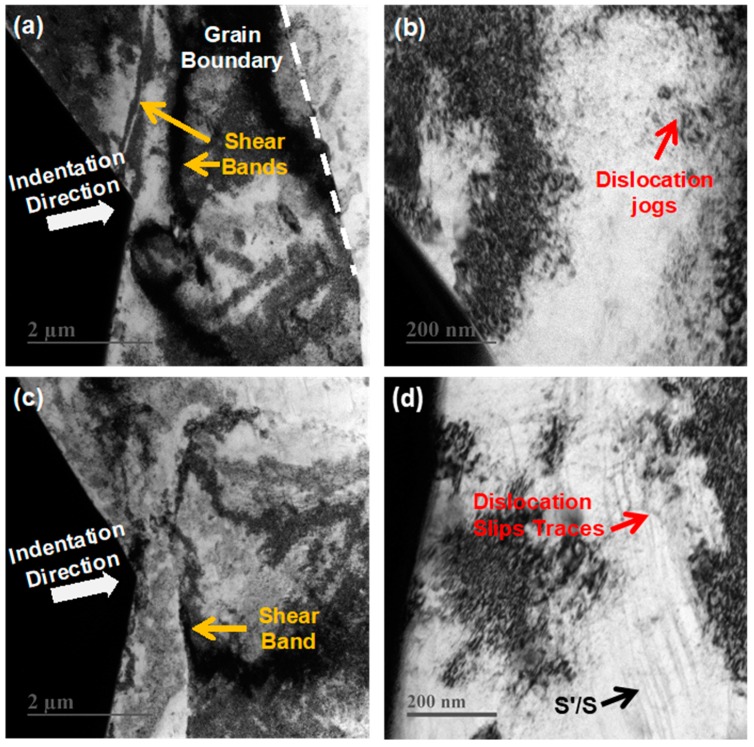
TEM micrographs of lamellae extracted from the indentation imprints of AA2524. (**a**,**b**) Pre-aged 0 h specimen and local magnification; (**c**,**d**) Pre-aged 12 h specimen and local magnification.

**Figure 9 materials-12-02907-f009:**
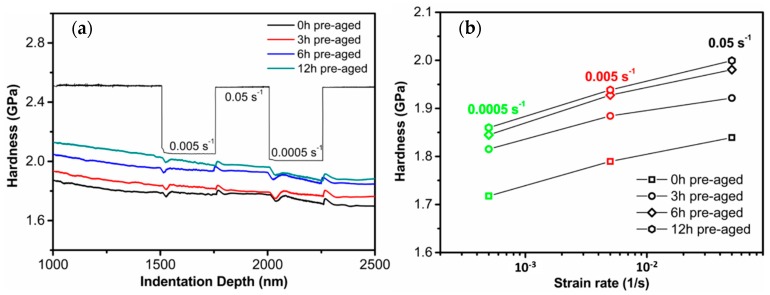
The indentation strain–rate jump experiment of pre-aged AA2524 with three different applied strain rates (**a**) and the correspondent results for hardness (**b**).

**Table 1 materials-12-02907-t001:** Chemical composition of the AA2524 alloy (wt%).

Zn	Mg	Cu	Mn	Si	Fe	Ti	Cr	Al
0.01	1.38	4.4	0.66	0.03	0.05	0.03	0.01	Bal.

**Table 2 materials-12-02907-t002:** Mechanical properties of the initial material with various pre-aging treatments.

Pre-Aging	Yield Strength/MPa	Tensile Strength/MPa	Elongation/%	Ultimate Creep Strain	Steady State Creep Rate (s^−1^)
0 h	265	403	22.18	0.00179	1.98 × 10^−6^
3 h	291	435	17.85	0.0015	1.91 × 10^−6^
6 h	303	440	13.64	0.00127	1.97 × 10^−6^
12 h	331	439	10.28	0.00119	1.88 × 10^−6^

**Table 3 materials-12-02907-t003:** Strain rate sensitivity (m) and activation volume (V*) of the creep aged AA2524 with different pre-aging treatments. B is the burger’s vector of pure Al, ≈0.286 nm.

Pre-Aged	m	V*(b^3^)
0 h	0.0186 ± 0.0021	27.50 ± 0.94
3 h	0.0182 ± 0.0009	26.67 ± 0.52
6 h	0.0169 ± 0.0018	28.18 ± 1.01
12 h	0.0171 ± 0.0016	27.08 ± 0.98
AVERAGE ± STDEV	0.0177 ± 0.0008	27.36 ± 0.64

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
