# Peer review of "Creep Mechanisms of an Al–Cu–Mg Alloy at the Macro- and Micro-Scale: Effect of the S′/S Precipitate"

_materials, 2019, doi:10.3390/ma12182907_

Round 1
Reviewer 1 Report
Abstract: please add the tensile creep test condition also in the abstract
line 67 : please add the thermomechanical process state of the commercial bar: it is a rolled-forged ...bar?
line 68: add: "room temperature 1-2% pre-deformation after solution…"
line 81: please insert the drawing of the specimens: it is not sufficient to put a reference for the reader. Please, specify that the macro creep tests are under tensile state and the micro under compression...or not?
line 177 and following: please specify how many test have been conducted per condition. This is need for a statistical assessment of the experimental error.
In fact your results are so close: n=2.41 - 2.50 that must be based on a strong statistic number of repetition-tests.
Fig 6: according to my experience the two series reported of tests can be confused as belonging to the same family.
This is the main point: the differences measured in the n-exponent are so close, that can be the result of experimental values spread and it is suggested to add more details about the number of tests per condition in order to have mean-values to be compared.
Author Response
Dear Editor and Reviewers:
Thank you for the detailed and helpful comments. We have revised the manuscript in accordance with your comments.
Reviewer1
Comments 1#
Abstract: please add the tensile creep test condition also in the abstract
line 67: please add the thermomechanical process state of the commercial bar: it is a rolled-forged ...bar?
line 68: add: "room temperature 1-2% pre-deformation after solution…"
Response 1#
The tensile creep test condition, the thermomechanical process state of the commercial bar and the condition of room temperature have been added in the revised manuscript as follows: “…, as revealed by the macro tensile creep tests at 180 ℃ and 180 MPa.”.
Comments 2#
line 81: please insert the drawing of the specimens: it is not sufficient to put a reference for the reader. Please, specify that the macro creep tests are under tensile state and the micro under compression...or not?
Response 2#
The geometry and dimensions of the creep specimen (Fig. 1) has been added. The describe of macro tensile creep tests has been improved.
Comments 3#
line 177 and following: please specify how many test have been conducted per condition. This is need for a statistical assessment of the experimental error. In fact your results are so close: n=2.41 - 2.50 that must be based on a strong statistic number of repetition-tests.
Response 3#
More than two repetitive tests have been conducted per condition for both macro and micro creep tests. The exponent coefficient n reported in the text was the average statistic number of repetition-tests.
Comments 4#
Fig 6: according to my experience the two series reported of tests can be confused as belonging to the same family.
This is the main point: the differences measured in the n-exponent are so close, that can be the result of experimental values spread and it is suggested to add more details about the number of tests per condition in order to have mean-values to be compared.
Response 4#
The n-exponent have been improved, as shown in Fig. 7 in revised manuscript. The error bars of the data have been added and the standard deviation of the n-exponent has been given in the text.
Reviewer 2 Report
- For the people that are not specialists is recommended to insert some explanations regarding S and S'. It due the fact that S and S' precipitates needs other temperature for formation (see: Acta Materials, 55 (2007) pp.933-941 "Two types of S phase precipitates in Al-Cu-Mg alloys). - Apart of this,in spite of the fact that in title is declares:"...effect of S/S' precipitation size..." this aspect not have special treatment in text. - Explain please if, in the fig.2, it is spoken about the same optical area before and after aging. If the answer is not, please offer a statistic appreciation of the grain size to have the right to state "...similary equiaxed grain structure for (a) 0h and (b) 12h pre-aged AA2524 alloy..." - In Tab 3. Both coefficients "m" and "V*(b3)" presented one anomaly at 6h aging time, in comparison with general tendency of decreasing with pre-aging time increasing. It needs some explanation and comments. The same anomaly is presented also in Tab.2, also at 6h pre-aging for steady-state creep rate (s-1) also without comments and explanations. - On the line 152: "...that prohibits effectively the dislocation activities...". On the line 257: "...however the steady state creep rate was approximately unchanged..." This final statement is based on the values 2.4...2.5 of the stress exponent,that imply macro steady creep mechanism dominated by grain boundaries sliding and dislocation slipping. - Comment: If at macro scale the steady state remain unchanged, all of research with different pre-aging duration remain without object. To understand the reason of this experimental research there are necessary more explanations and justifications
Author Response
Response to the Editor and Reviewers
Dear Editor and Reviewers:
Thank you for the detailed and helpful comments. We have revised the manuscript in accordance with your comments.
Reviewer2
Comments 1#
- For the people that are not specialists is recommended to insert some explanations regarding S and S'. It due the fact that S and S' precipitates needs other temperature for formation (see: Acta Materials, 55 (2007) pp.933-941 "Two types of S phase precipitates in Al-Cu-Mg alloys). - Apart of this, in spite of the fact that in title is declares:"...effect of S/S' precipitation size..." this aspect not has special treatment in text.
Response 1#
Thanks for the kind suggestion. Some explanations regarding S and S' precipitates have been added in the revised manuscript. In addition, the title of the manuscript has revised as “Creep mechanisms of Al-Cu-Mg alloy at macro and micro scales: effect of S'/S precipitate”.
Comments 2#
- Explain please if, in the Fig.2, it is spoken about the same optical area before and after aging. If the answer is not, please offer a statistic appreciation of the grain size to have the right to state "...similary equiaxed grain structure for (a) 0h and (b) 12h pre-aged AA2524 alloy..."
Response 2#
The optical areas before and after aging come from different specimens of AA2524. Therefore, grain size statistics was carried out using Image-Pro software. More than 2000 grains were measured for 0h and 12h pre-aged AA2524. The average grain size has been given in the revised manuscript.
Comments 3#
- In Tab 3. Both coefficients "m" and "V*(b3)" presented one anomaly at 6h aging time, in comparison with general tendency of decreasing with pre-aging time increasing. It needs some explanation and comments. The same anomaly is presented also in Tab.2, also at 6h pre-aging for steady-state creep rate (s-1) also without comments and explanations.
Response 3#
The experimental results show that the change of m and V* with nucleation and growth of S/S' precipitates is not obvious, because the micro creep mechanism is mainly caused by dislocation interaction (TEM photos shown in Fig. 8). Considering the error of experimental data, the result of 6h pre-aged is close to other results.
Comments 4#
- On the line 152: "...that prohibits effectively the dislocation activities...".
On the line 257: "...however the steady state creep rate was approximately unchanged..." This final statement is based on the values 2.4...2.5 of the stress exponent,that imply macro steady creep mechanism dominated by grain boundaries sliding and dislocation slipping.- Comment: If at macro scale the steady state remain unchanged, all of research with different pre-aging duration remain without object. To understand the reason of this experimental research there are necessary more explanations and justifications
Response 4#
In this study, we focused on the effect of aging precipitates on the creep mechanism. Although the macro creep mechanism has no change under different pre-aged AA2524, the contribution of aging precipitates cannot be explained directly because the macro creep deformation includes dislocation, aging precipitates, grain boundary and defects. Therefore, we use micro creep method to eliminate the contribution of grain boundary and defects to creep and study the effect of dislocation and precipitation on relative creep in a micro-scale. At the same time, with TEM, we found that the creep in micro-scale is mainly controlled by dislocation. Therefore, combining macro-micro creep, we concluded that aging precipitation has no obvious contribution to creep.
Round 2
Reviewer 1 Report
Line 192 - 195: Your experiments are confirming that the "n" exponent is not affected by the different pre-aging treatments. In fact, the error data dispersion does not permit to identify any specific difference. The fact that there is a heavy n-value overlap suggests to state clearly that the n exponent it is not affect by the pre-aging treatments.
line 242: ...0.0169-0.0186 instead of 0.0171- 0.0186
Table 3: Please, give some some comment about the m value at 6hours and 12 hours. Explain the fact that m remains almost constat....
Author Response
Dear Editor and Reviewers:
Thank you for the detailed and helpful comments. We have revised the manuscript in accordance with your comments.
Comments 1#
- Line 192 - 195: Your experiments are confirming that the "n" exponent is not affected by the different pre-aging treatments. In fact, the error data dispersion does not permit to identify any specific difference. The fact that there is a heavy n-value overlap suggests to state clearly that the n exponent it is not affect by the pre-aging treatments.
Response 1#
Thanks for the kind suggestion. We have corrected the related discussion in the revised version, i.e., “Considering the error tolerance with heavy n-value overlap, the result suggests strongly that the n exponent is independent of the pre-aging treatments”.
Comments 2#
- line 242: ...0.0169-0.0186 instead of 0.0171- 0.0186
Response 2#
The data has been corrected.
Comments 3#
- Table 3: Please, give some some comment about the m value at 6hours and 12 hours. Explain the fact that m remains almost constant.....
Response 3#
More details in m and V* value describes have been added. Also, the average and standard deviation of the m (0.0177±0.0008) and V*(≈27.36±0.64 b3) of four different preaged samples are reported in Table 3. The low standard deviation of the data indicates that the strain rate sensitivities and activation volumes remain almost constant.
Reviewer 2 Report
Is good that you change the title of the paper so to be suitable with content;
Also is good that you clarify the problem of S'/S aligning this at the paper experiment not more;
Please insert in the paper some considerations regarding the differences mentioned in my revision on the experimental data, at least at the level of the comment which you send to me;
Author Response
Dear Editor and Reviewers:
Thank you for the detailed and helpful comments. We have revised the manuscript in accordance with your comments.
Comments 1#
- Is good that you change the title of the paper so to be suitable with content;Also is good that you clarify the problem of S'/S aligning this at the paper experiment not more; Please insert in the paper some considerations regarding the differences mentioned in my revision on the experimental data, at least at the level of the comment which you send to me;
Response 1#
Thanks for the kind suggestion. More details in m and V* value describes have been added. Also, the average and standard deviation of the m (0.0177±0.0008) and V*(≈27.36±0.64 b3) of four different preaged samples are reported in Table 3. The low standard deviation of the data indicates that the strain rate sensitivities and activation volumes remain almost constant.